# CONSISTENCY-BASED ANOMALY DETECTION WITH ADAPTIVE MULTIPLE-HYPOTHESES PREDICTIONS

## ABSTRACT

In one-class-learning tasks, only the normal case can be modeled with data, whereas the variation of all possible anomalies is too large to be described sufficiently by samples. Thus, due to the lack of representative data, the wide-spread discriminative approaches cannot cover such learning tasks, and rather generative models, which attempt to learn the input density of the normal cases, are used. However, generative models suffer from a large input dimensionality (as in images) and are typically inefficient learners. We propose to learn the data distribution more efficiently with a multi-hypotheses autoencoder. Moreover, the model is criticized by a discriminator, which prevents artificial data modes not supported by data, and which enforces diversity across hypotheses. This consistency-based anomaly detection (ConAD) framework allows the reliable identification of out-of-distribution samples. For anomaly detection on CIFAR-10, it yields up to 3.9% points improvement over previously reported results. On a real anomaly detection task, the approach reduces the error of the baseline models from 6.8% to 1.5%.

## 1 INTRODUCTION

Anomaly detection classifies a sample as normal or abnormal. In many applications, however, it must be treated as a one-class-learning problem, since the abnormal class cannot be defined sufficiently by samples. Samples of the abnormal class can be extremely rare, or they do not cover the full space of possible anomalies. For instance, in an autonomous driving system, we may have a test case with a bear or a kangaroo on the road. For defect detection in manufacturing, new, unknown production anomalies due to critical changes in the production environment can appear. In medical data analysis, there can be unknown deviations from the healthy state. In all these cases, the well-studied discriminative models, where decision boundaries of classifiers are learned from training samples of all classes, cannot be applied. The decision boundary learning of discriminative models will be dominated by the normal class, which will negatively influence the classification performance.

Anomaly detection as one-class learning is typically approached by generative, reconstruction-based methods (Zong et al., 2018) . They approximate the input distribution of the normal cases by parametric models, which allow them to reconstruct input samples from this distribution. At test time, the data log-likelihood serves as an anomaly-score. In the case of high-dimensional inputs, such as images, learning a representative distribution model of the normal class is hard and requires many samples.

Typically, an autoencoder-based approach such as the variational autoencoder (Rezende et al., 2014; Kingma & Welling, 2013) is used. Autoencoders tend to produce blurry reconstructions, since they regress the conditional mean, and cannot model multi-modal distributions; see Fig. 1 for an example on a Metal Anomaly dataset. Due to multiple modes in the actual distribution, the approximation with the mean predicts high probabilities in areas not supported by samples. The blurry reconstructions in Fig. 1 should have a low probability and be classified as anomalies, but they have the highest likelihood under the learned autoencoder.

Multiple-hypotheses networks could give the model more expressive power Rupprecht et al. (2016a), Chen & Koltun (2017), Ilg et al. (2018), Bhattacharyya et al. (2018). In conjunction with autoencoders, the multiple hypotheses can be realized with a multi-headed decoder. Concretely, each network head may predict a Gaussian density estimate.

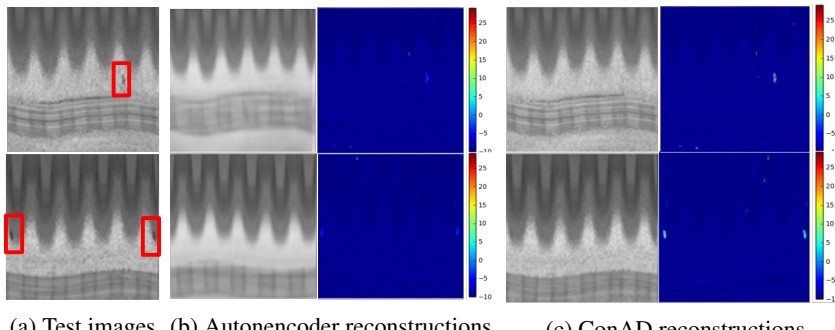

(a) Test images    (b) Autonencoder reconstructions      (c) ConAD reconstructions

Figure 1: Detection of anomalies on a Metal Anomaly dataset. (a) Test images showing anomalies (black spots). (b) An Autoencoder-based approach produces blurry reconstructions to express model uncertainty (c) Our model: Consistency-based anomaly detection (ConAD) gives the network more expressive power with a multi-headed decoder (also known as multiple-hypotheses networks). The resulting anomaly scores are hence much clearer in our framework ConAD.

Multiple-hypotheses networks were not yet applied to anomaly detection due to several difficulties in training these networks to produce a multi-modal distribution consistent with the training distribution. The loosely coupled hypotheses branches are typically learned with a winner-takes-all loss, where all learning signal is transferred to one single best branch. Hence, bad hypotheses branches are not penalized and may support non-existing data regions. The artificial data modes, therefore, cannot be distinguished from normal data. This is an undesired property for anomaly detection and becomes more severe with an increasing number of hypotheses. Furthermore, the majority of multiple-hypotheses-branches tend to concentrate on the most dominant data modes. This hypotheses concentration leads to over-fitting in the neighborhood of dominant modes and under-fitting in underrepresented data regions. This, too, has a negative effect on the estimated anomaly scores.

Alternatively, mixture density networks (MDNs) (Bishop, 1994) provide a strict coupling of hypotheses branches. These models learn a conditional Gaussian mixture distribution. Hence, the hypotheses are coupled via mixing coefficients into a single likelihood function. Anomaly scores for new points can be estimated using the data likelihood, as formally defined in Appendix A.

Fig. 2 illustrates the different strategies. A single-mode autoencoder (b) fails in case of multi-modal distributions. MDNs (c) in principle can be used for abnormality detection even for multimodal distributions. However, global, multi-modal distribution estimation is a hard learning problem that does not work as perfectly in practice as shown in this illustration. For instance, MDNs tend to suffer from mode collapse in high-dimensional data spaces, i.e., the relevant data modes needed to distinguish rare but normal data from anomalies will be missed. Contrary, Local-outlier-factor operates in images-space directly without training which (1) fails in very high-dimensional spaces (2) is slow at test time.

In this work, we adopt multiple-hypotheses networks for anomaly detection to provide a more fine-grained description of the data distribution than a single-headed network. Hypotheses are meant to form clusters in the data space and can capture model uncertainty not encoded by the latent code. We reduce the problem of artificial data modes by combining multiple-hypotheses learning with a discriminator D as a critic. The discriminator ensures the consistency of estimated data modes w.r.t. the real data distribution.

Moreover, we propose to focus on the local neighborhood and to estimate the fit of a sample to the distribution model based on the distance to the closest cluster. This avoids issues with global distribution estimation methods, such as mode collapse. Hypotheses rather act as local, single mode density estimates and are easier and more sample-efficient to learn than a full multi-modal distribution. Fig. 3c shows our framework applied to a variational autoencoder.

We evaluate anomaly detection performance of our approach on CIFAR-10 and a real anomaly image dataset, the "Metal Anomaly dataset" with images showing a structured metal surface, where anomalies in the form of scratches, dents or texture differences are to be detected. We show that anomaly detection performance with multiple-hypotheses networks is significantly better compared

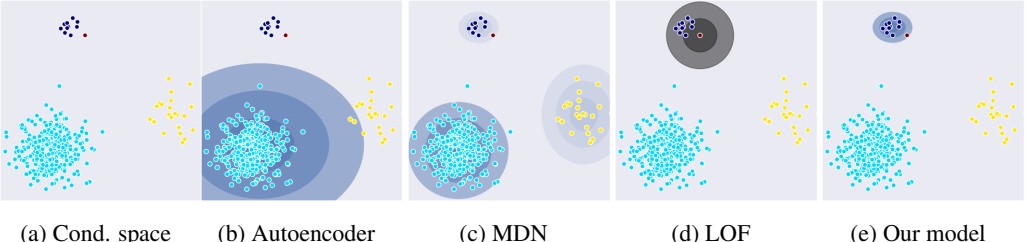

|  (a) Cond. space | (b) Autoencoder | (c) MDN | (d) LOF | (e) Our model |

Figure 2: Local and global neighborhood-based anomaly detection: Here, two pixel dimensions (a) with details that are hard to capture in the conditional space are shown. The red dot is a new point. Dark blue indicates high likelihood, black indicates the neighborhood considered. The autoencoder (b) cannot deal with the multi-modal distribution. The mixture density network (c) in principle can do so, but recognition of the sample as a normal case is very brittle and will fail in case of mode collapse. In contrast, Local-Outlier-Factor (d) and our model (e) consider only the local neighborhood for anomaly score estimation and more reliably classify the point. In our model, we encourage multiple hypotheses to cover different modes. In each hypothesis branch, the probability mass is distributed only within the cluster and not beyond.

to single-hypotheses networks. On CIFAR-10, our proposed ConAD framework (consistency-based anomaly detection) improves on previously published results. Furthermore, we show a large performance gap between ConAD and Mixture Density networks (MDNs). This indicates that anomaly score estimation based on the global neighborhood (or data likelihood) is inferior to local neighborhood consideration.

## 2 ONE-CLASS LEARNING FOR ANOMALY DETECTION

Traditional one-class learning techniques (Schölkopf et al., 2001; Tax & Duin, 2004; Liu et al., 2008; 2012; Breunig et al., 2000) often fail in high-dimensional input domain and require careful features selection (Zong et al., 2018) . To cope with high-dimensional domains, typically a reconstruction-based approach is used. This paradigm comprises two steps: (1) during training, learn the normal data distribution and (2) at test time, use the negative likelihood for contaminated data as their anomaly score.

Recently, advances in generative modeling such as Generative Adversarial Network (GAN) (Goodfellow et al., 2014) and Variational Autoencoder (VAE) (Rezende et al., 2014; Kingma & Welling, 2013) are used for anomaly detection (Zong et al., 2018; Schlegl et al., 2017; Deecke et al., 2018). However, GAN and VAE approaches have limitations in anomaly detection tasks. The GAN tends to assign less probability mass to real samples while VAE typically regress to the conditional means, which can be seen from the blurry reconstructions. The mean regression in VAE express the model uncertainty and falsify the reconstruction-errors for unseen images.

One simple way to address model uncertainty in VAE is giving the decoder additional expressive power with multi-headed decoders. The idea is to approximate multiple conditional modes (dense data regions) by using multiple headed networks. This idea leads to training of multiple networks in Multi-Choice-learning (Dey et al., 2015; Lee et al., 2017; 2016), the estimation of conditional Gaussian Mixture model in Mixture Density Network (MDN) (Bishop, 1994) and multiple-hypotheses predictions (MHP) (Ilg et al., 2018; Chen & Koltun, 2017; Bhattacharyya et al., 2018; Rupprecht et al., 2016a). In MDN, the mixtures are strictly coupled via mixture coefficients while mixtures in MHPs act as loosely coupled local density estimators. In MHP, only the best hypothesis branch will receive a learning signal, that is, if it makes the closest guess to the training sample.

For anomaly detection, our model uses MHP-training with VAE to address the model uncertainty directly. In MDN, the anomaly score is proportional to weighted distances to all data modes and in MHP only to closest data mode. To highlight the change in paradigm, we refer to this learning in MHP as consistency-based learning. Samples have a small effect on the loss as long they are close to one single data mode. The learning dynamic in MHP is also different and more efficient than

(a) Single-headed networks (b) Multi-headed networks  (c) Multi-headed network with discriminator training

Figure 3: Illustration of multiple-hypotheses networks compared to single-hyptohesis network.. Our ConAD framework (c), which integrates a discriminator D to avoid support of non-realistic data modes and foster higher mode coverage with the generated hypotheses.

in MDN: the number of samples with a high loss is lower. In this context, we relax the learning objective from density-based to consistency-based learning.

In Local Outlier Factor (LOF) (Breunig et al., 2000), the outlier-score only depends on the local neighborhood. The outlier score proportional to the mean density of neighboring points divided by the local point density. Hence, samples further away do not influence the outlier-score. Motivated by this heuristic, our model employs learning of many loosely decoupled local density estimates with MHP-learning. Our model (1) concentrates only on the closest data mode instead of considering the data likelihood for outlier detection (2) and enables easier learning due to consistency-based learning instead of full density estimation. LOF computes the outlierness only on test-time and in input spaces directly. Contrary, our model first approximate the data manifold and subsequently performs anomaly detection in the input space under the learned model.

The MHP-technique has been used for uncertain tasks such as future prediction (Rupprecht et al., 2016b) or optical flow prediction (Ilg et al., 2018). In the simplest form, the multiple networks heads learn from a winner-takes-all (WTA) loss, whereby only the best branch receives the learning signal. Previous works employ loss extension such as the use of a smoothing loss (Ilg et al., 2018) or distribution of learning signal to non-optimal branches (Rupprecht et al., 2016b) to generate diverse and meaningful hypotheses.

Compared to our framework, previous MHP-approaches were not developed for distribution learning. There is no explicit mechanism to avoid mode collapse among hypotheses. Furthermore, generated hypotheses could support non-existing data regions, which can be fatal for anomaly detection tasks. Contrary, our framework ConAD employs a discriminator D to assess the quality of the generated hypotheses and to avoid support of non-existent data modes. To reduce hypotheses mode collapse, our model employs hypotheses discrimination. In the spirit of minibatch discrimination (Salimans et al., 2016), D additionally receives pair-wise distances across a batch of hypotheses. Since a batch of real samples is typically diverse, D can detect a homogeneous batch of hypotheses as fake easily.

# 3 LEARNING WITH MULTIPLE-HYPOTHESES-PROPOSALS (MHP) NETWORKS FOR ANOMALY DETECTION

Typically in distribution learning, Autoencoder-approaches regress the means and produce blurry reconstructions. Therefore, we propose to employ MHP as additional expressive power for the decoder (Fig 3 (a-b)). First, we discuss two possible shortcomings of multiple-hypotheses learning: support of artificial data mode and hypotheses mode collapse. Subsequently, we show how to reduce these effects with discriminator training and hypotheses discrimination (Fig 3 c).

### 3.1 Shortcoming of multiple-hypotheses learning

**Support of artificial data mode in one-to-many mapping tasks**   To understand the shortcomings of learning with multiple-hypotheses-proposals (MHP), first consider a simple one-to-many mapping task from $x$ to $y$ as given in Fig. 4. Unimodal models (i.e., single-headed networks) fail to capture to data distribution.

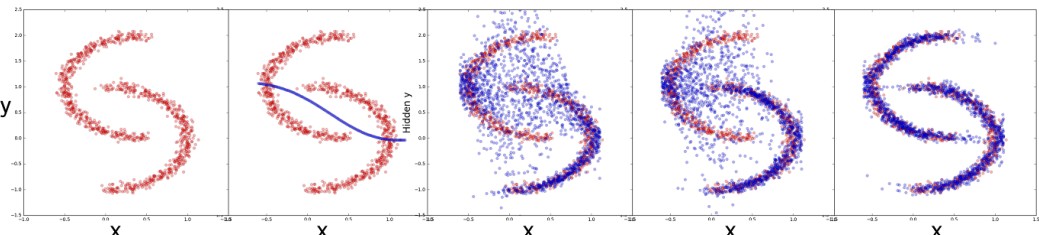

Figure 4: Flipped half-moon data-set: mapping from $x$ to $y$ is not unique, e.g. for $x = 0$, there are four different modes. Left to right: with an increasing number of mixture components in a mixture density network, the data distribution can be modeled increasingly well.

Similar to Mixture Density networks, each hypothesis branch in MHP-networks represents a Gaussian density function with a mean and variance. Typically, MHP-networks learns from the winner-takes-all (MHP-WTA) loss in Eq. 1:

$$L_{WTA}(y|x) = E_{x_i}\left[\log p_{\theta_h}(y|x_i)\right] \text{ s.t. } h = \arg\max_j E_{x_i}\left[\log p_{\theta_j}(y|x_i)\right] \qquad (1)$$

Whereby $\theta_j$ is the parameter set of hypothesis branch $j$, $\theta_h$ the best hypothesis concerning data likelihood given a sample $x_i$. In other words, only the network head with the best-matching hypothesis concerning the training samples receives the learning signal. The best hypothesis is the one with the highest sample likelihood (or minimal distance to sample if the variance is equal for all hypotheses). Additionally, Rupprecht et al. (2016a) proposed a $\epsilon$-smoothed loss. With this loss, a small $\epsilon$-ratio of the learning signal is distributed among non-optimal hypotheses branches. We refer to this loss as learning with MHP-loss (Appendix 11).

However, learning with MHP or MHP-WTA may result in support of artificial (non-existing) data modes. Fig. 5 illustrates this problem, which we refer to as inconsistency concerning the underlying distribution. In regions where the half-moon abruptly ends, the hypotheses (in MHP and MHP-WTA) continue and support non-existing data regions. This inconsistency effect is fatal for anomaly detection. More details can be found in the experiments on the toy dataset in the appendix B. Intuitively, in learning with the winner-takes-all loss, the non-optimal hypotheses are not penalized.

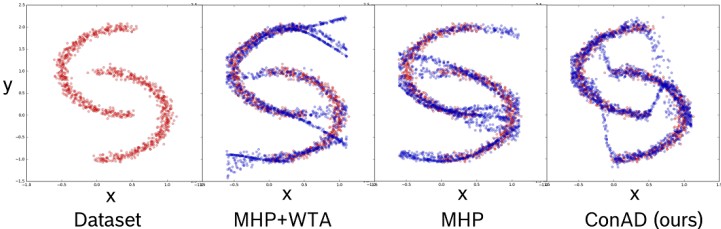

Figure 5: Flipped half-moon dataset: conditional prediction of $y$ based on $x$. Red points are samples from true distribution while blue points represent samples from distributions approximations. Learning with multiple-hypotheses predictions (MHP) loss or MHP + Winner-takes-all (WTA) loss lead to support of artificial data regions. Our approach ConAD reduces this effect.

Therefore they can support artificial data regions without being informed via the learning signal. A more formal discussion can be found in Appendix D.

The learning signal distribution with $\epsilon$-parameter attempts to reduce support of artificial regions. However, finding good $\epsilon$ is crucial and difficult. If $\epsilon = 0$, the MHP loss corresponds to learning with MHP-WTA. If $\epsilon = \frac{H-1}{H}$, whereby $H$ is the number of hypotheses branches, all hypotheses will regress to the same conditional mean. A more formal discussion can be found in Appendix E. Additionally, $\epsilon$ is an additional hyper-parameter to be chosen. Choosing proper hyper-parameters in one-class-learning is difficult since there is no anomaly available at training time.

**Distribution learning with Autoencoder as a one-to-many mapping task**   Training Autoencoders with likelihood-metric often results in blurry reconstructions. This blurriness is fatal for anomaly detection since it falsifies the reconstructions error. This effect can be understood as a regression to the conditional mean. That means, after training convergence, each point on the learned manifold still represents many different data points in the input space. In other words, the mapping from latent code to input space is a one-to-many mapping.

Certainly, in the optimal training case, each point on the data manifold should represent one single input vector. However, this optimality requires either significantly more data to reduce the model uncertainty or powerful encoder network and latent code or both. Contrary, we propose to let the Autoencoder express the model uncertainty with the multiple-hypotheses directly. Hence, the change to Autoencoder is very simple, and no more data is required than before.

**Mode collapse across hypotheses**   Furthermore, with the MHP and MHP-WTA learning objective, the hypotheses are encouraged to cover the existing modes. When there are more hypotheses available than data modes, most of the hypotheses will tend to concentrate on the most dominant data modes. This mode collapse can be avoided by enforcing diversity across hypotheses, which is similar to maximizing inter-class variance across clusters defined by the hypotheses.

### 3.2   Consistency-based anomaly detection (ConAD) with MHP and discriminator D

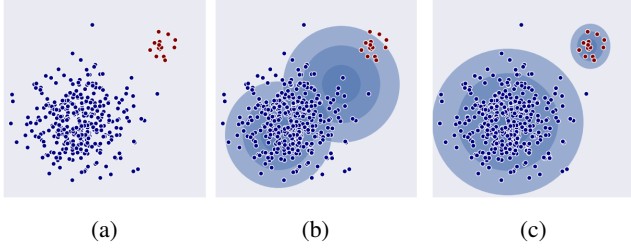

(a)                              (b)                              (c)

Figure 6: (a) shows a modeling task with one extremely dominant data mode (dense region) and one under-represented mode. (b) shows how multiple-hypotheses predictions are used to cover data modes. Hypotheses tend to concentrate on dominant mode, which leads to over-fitting in this region. (c) Increasing diversity across hypotheses (similar to maximizing inter-class variance) leads to better clusters

We propose multiple-hypotheses Variational Autoencoder (VAE) for learning the normal data distribution for anomaly detection tasks. Each hypothesis branch can be seen as a cluster in the data conditional space. Anomalies are detected using the distance to next local clusters, in contrast to distances to all clusters in Mixture Density networks (MDN) (Bishop, 1994). To avoid coverage of non-existing data regions by the hypotheses, we propose to use a discriminator as a critic. Further, we employ hypothesis discrimination to encourage diversity among hypotheses. This constraint is similar to the improvement of inter-class variance among clusters. The details are explained in the following.

**Learning with multiple-hypotheses predictions (MHP) in Variational Autoencoder**   In this work, we consider distribution learning in an Autoencoder as a one-to-many-mapping. We propose to let the network express the model uncertainty in the conditional input space with multiple hypotheses predictions (MHP). The hypotheses can be seen as a set of local density estimates (or

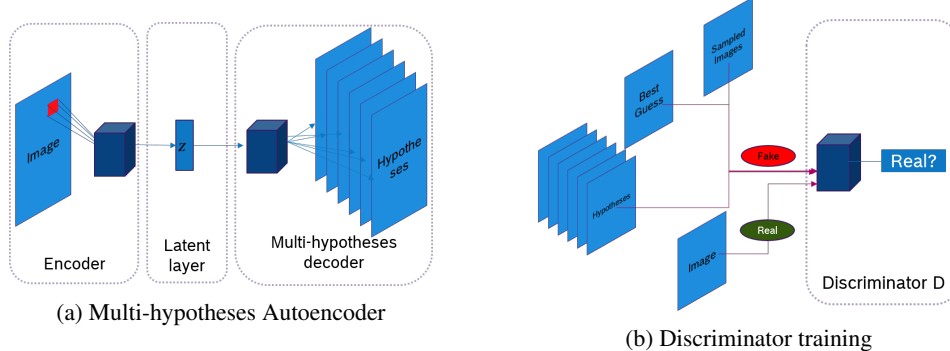

(a) Multi-hypotheses Autoencoder

(b) Discriminator training

Figure 7: ConAD: our multiple-hypotheses autoencoder and with the training discriminator training.

cluster). In contrast to that, Mixture Density Network (MDN) predicts a Gaussian Mixture model in the conditional space. We refer to this estimate as a global density estimate.

The learning of different hypotheses is performed based on a winner-takes-all-objective as given in Eq. 2.

$$L_{WTA}(x) = E_{z_i \sim q_\phi(z_i|x)} \left[ \log p_{\theta_h}(x|z_i) \right] \text{ s.t. } h = \arg\max_j E_{z_i \sim q_\phi(z_i|x)} \left[ \log p_{\theta_j}(x|z_i) \right] \quad (2)$$

Whereby $L_{WTA}$ is the winner-takes-all energy function, $1 \leq j \leq H$ indicates the different hypotheses networks, $z_i$ the respective latent code. To reduce free parameters, hypotheses networks with params $\theta_j$ share all layers but the last output layer. Intuitive, it means that only the best matching hypothesis receives all of the learning signals from the negative log-likelihood (NLL) loss during training.

An efficient variant to realize MHP in neural networks is by using multi-headed-networks. In this variant, only the last layer is split to provide different hypotheses. All other layers are shared as shown in Fig. 3c. Our framework is based on the Variational Autoencoder (Kingma & Welling, 2013; Rezende et al., 2014) which provides an effective manifold learning and an efficient inference stage with a parameterized encoder $q_\phi$.

**Discriminator D to avoid non-existent mode coverage and mode collapse of hypotheses**   Hypotheses generated by the MHP-networks could support artificial data regions not covered by real samples due to the WTA loss. To alleviate this, we propose to match the density estimates with MHP to the real underlying density. The auxiliary task is to learn from a symmetric variant of the Kullback-Leibler divergence (KLD). In detail, we employ the Jensen-Shannon divergence (JSD)-metric by using discriminator D as a critic for generated hypotheses. Fig. 3c illustrates a sample realization with VAE.

More concretely, the D and G are in a mini-max game in Eq. 3.

$$\min_D \max_G L_D(x, z) = \min_D \max_G \underbrace{-\log(p_D(x_{real}))}_{L_{real}} + L_{fake}(x, z) \quad (3)$$

$$L_{fake}(x, z) = \log(p_D(\hat{x}_{z \sim \mathcal{N}(0,1)})) + \log(p_D(\hat{x}_{z \sim \mathcal{N}(\mu_{z|x}, \Sigma_{z|x})})) + \log(p_D(\hat{x}_{\text{best\_guess}})) \quad (4)$$

In this energy formulation, the standard GAN loss is extended to assure the quality of generated hypotheses. Figure 7 illustrates how samples are fed into the discriminator. Samples labeled as fake are: randomly-sampled images $\hat{x}_{z \sim \mathcal{N}(0,1)}$, data reconstruction defined by individual hypotheses $\hat{x}_{z \sim \mathcal{N}(\mu_{z|x}, \Sigma_{z|x})}$, the best combination of hypotheses according to the Winner-takes-all-loss $\hat{x}_{\text{best\_guess}}$.

Accordingly, the learning objective for the VAE generator becomes:

$$\min_G L_G = \min_G L_{WTA} + KLD(q_\phi(z|x)||\mathcal{N}(0,1)) - L_D \quad (5)$$

| Name | Problem | Tasks | Resolution | Normal data | | | Anomaly |
| --- | --- | --- | --- | --- | --- | --- | --- |
| | | | | Train | Valid | Test | Test |
| CIFAR-10 | 1 vs. 9 | 10 | 32x32 | 4500 | 500 | 1000 | 9000 |
| Metal anomaly | 1 vs. 1 | 1 | 224x224 | 5408 | 1352 | 1324 | 346 |

Table 1: Dataset description. Cifar-10 is transformed into 10 anomaly detection tasks, whereby one class is used as the normal class, the remaining classes are the anomalies. Further, note that the train & validation dataset contains only normal data samples. This scenarios resembles the typical situations where anomalies are extremely rare and not available at training time.

To address the mode collapse problem of hypotheses, we propose to employ hypotheses discrimination (based on minibatch discrimination (Salimans et al., 2016)). In each batch, the discriminator receives the pair-wise features distance across generated hypotheses. Since batches of real images have large pair-wise distances, the generator has to generate diverse outputs to avoid being detected too easily.

In summary, our framework ConAD proposes multiple-hypotheses learning with a VAE, supported by a discriminator D to avoid support of non-existing data modes and foster mode coverage. The local likelihood estimates given by the closest hypothesis are used for anomaly detection.

## 4 EXPERIMENTS

### 4.1 EXPERIMENTS DESCRIPTIONS

In this section, we focus on the evaluation of our approach compared to recent deep learning and non-deep learning techniques for one-class learning tasks. In these tasks, anomalies are extremely rare and hence not available at training time. The main effort comes from the collection of a large dataset to receive anomalies, not from the labeling activity.

The details of the proposed framework; consistency-based anomaly detection (ConAD) is explained in the following. A Variational Autoencoder Kingma & Welling (2013) with Gaussian output distribution is employed as a baseline model. The decoder is then extended to a multiple-head-network to support multiple-hypotheses. Each hypothesis itself predicts a Gaussian density estimate. The outputs from the Autoencoders are criticized by a discriminator D. The network architecture follows principles from Radford et al. (2015b) and Springenberg (2015). Fig. 3 c) shows such a network conceptually. The framework can be easily extended to recent advances in deep generative modeling. Quantitative evaluation is done on CIFAR-10 and the Metal Anomaly dataset. The typical 10-way classification task in CIFAR-10 is transformed into 10 one vs. nine anomaly detection tasks. Each class is used as the normal class once; all remaining classes are treated as anomalies. Details can be found in Tab. 1. During model training, only data from the normal data class is used, data from anomalous classes are abandoned. At test time, anomaly detection performance is measured in Area-Under-Curve of Receiver Operating Curve (AUROC) based on normalized negative log likelihood scores given by the training objective.

In Tab. 2, we evaluated on CIFAR-10 variants of our multiple-hypotheses approaches including the following energy formulations: MDN (Bishop, 1994), MHP-WTA (Ilg et al., 2018), MHP (Rupprecht et al., 2016a), ConAD, and MDN+ConAD. We compare our methods against vanilla VAE (Kingma & Welling, 2013; Rezende et al., 2014) , VAEGAN (Larsen et al., 2015; Dosovitskiy & Brox, 2016), AnoGAN (Schlegl et al., 2017), AdGAN Deecke et al., 2018, OC-Deep-SVDD (Ruff et al., 2018). Traditional approaches considered are: Isolation Forest (Liu et al., 2008; 2012), OCSVM (Schölkopf et al., 2001). The performance of traditional methods suffers due to the curse of dimensionality (Zong et al., 2018).

Furthermore, on the high-dimensional Metal anomaly dataset, we focus only on the evaluation of deep learning techniques. The GAN-techniques proposed by previous work AdGAN & AnoGAN heavily suffer from instability due to pure GAN-training on a small dataset. Hence, their training leads to random anomaly detection performance. Therefore, we only evaluate MHP-based approaches against their uni-modal counterparts (VAE, VAEGAN).

| | Traditional models | | | Deep Learning models | | |
|---|---|---|---|---|---|---|
| KDE-PCA | OC-SVM-PCA | IF | GMM | AnoGAN | ADGAN | OC-D-SVDD |
| .590 | .610 | .558 | .585 | .612 | .620 | .632 |

| | Multiple hypothesis models | | | | |
|---|---|---|---|---|---|
| Hypotheses branches | MHP | MHP+WTA | MDN | MDN+ConAD | ConAD |
| 1 | | .610 (= VAE) | | .609 (= VAE-GAN) | |
| 2 | .619 | .622 | .609 | .616 | **.643** |
| 4 | .619 | .622 | .610 | .621 | **.639** |
| 8 | .618 | .619 | .610 | .623 | **.671** |
| 16 | .617 | .620 | .609 | .614 | **.659** |

Table 2: Anomaly detection on CIFAR-10, performance measured in AUROC. Each class is considered as the normal class once with all other classes being considered as anomalies, resulting in 10 one-vs-nine classification tasks. Performance is averaged for all ten tasks and over three runs each. See appendix for detailed performance. Our approach significantly outperforms previous traditional and deep learning methods.

| | Multiple hypothesis models | | | | |
|---|---|---|---|---|---|
| Hypotheses branches | MHP | MHP+WTA | MDN | MDN+ConAD | ConAD |
| 1 | | .942 (= VAE) | | .936 (= VAE-GAN) | |
| 2 | .980 | .980 | .900 | .942 | **.985** |
| 4 | .970 | **.980** | .910 | .913 | .977 |
| 8 | .950 | .946 | .916 | .943 | **.965** |

Table 3: Anomaly detection performance on Metal Anomaly dataset. To reduce noisy residuals due to the high-dimensional input domain, only 10% of maximally abnormal pixels with the highest residuals are summed to form the total anomaly score. AUROC is computed on an unseen test set, a combination of normal and anomaly data. For more detailed results, refer to attachment H. Anomaly detection performance of plain MHP rapidly breaks down with increasing number of hypotheses.

## 4.2 CIFAR-10

Tab. 2 shows an extensive evaluation of different traditional and deep learning techniques. Results are adapted from Deecke et al. (2018) in which the training and testing scenarios were similar. Refer to Appendix. G for more results. Traditional, non-deep-learning methods only succeed to capture classes with a dominant homogeneous background such as ships, planes, frogs (backgrounds are water, sky, green nature respectively). This issue occurs due to preceding feature projection with PCA, which focuses on dominant axes with large variance. Deecke et al. (2018) reported that even discriminative features from a pretrained AlexNet have no positive effect on anomaly detection performance.

In contrast to that, deep learning methods are performing significantly better, even without careful parameter tuning. When the MHP-technique is applied to this task, a performance comparable to previously reported deep learning, but non-MHP results is achieved. Note that having the multiple output distributions is not sufficient to meet high performance: MDNs are performing worse than the local density estimation provided by the MHP-technique. Nevertheless, the best performance is achieved in our ConAD- framework, by utilizing the flexibility of multiple hypotheses more effectively, leading to significantly higher detection performance of up to 5.1% absolute improvement.

## 4.3 METAL ANOMALY DATASET

Tab.3 shows an evaluation of MHP-methods against density-learning methods such as VAE (Kingma & Welling, 2013), MDN (Bishop, 1994), VAEGAN (Dosovitskiy & Brox, 2016; Larsen et al., 2015). Note that the VAE-GAN model corresponds to our ConAD with a single hypothesis. The VAE corresponds to a single hypothesis variant of MHP, MHP-WTA, and MDN.

The significant improvement of up to 4.2% AUROC-score comes from our relaxation of density estimation into local density estimation in the spirit of LOF (Breunig et al., 2000), i.e., each dense data region (mode) receives at least one hypothesis to cover the local density. In a high-dimensional domain such as images, anomaly detection with MDN is worse than with our approach MHP approaches. Consider images with an extremely rare value in one pixel-dimension. The Mixture Density models evaluate likelihood based on all data modes found for this pixel. In contrast to that, MHP-models only considers which data mode is the closest and computes the local likelihood as the anomaly score. The local neighborhood suppresses the over-estimation of anomaly degree compared to a global likelihood.

Using the MHP-technique, better performance is already achieved with two hypotheses. However, without the discriminator D, an increasing number of hypotheses rapidly leads to performance breakdown, due to the inconsistency property of generated hypotheses as discussed earlier. Intuitively, additional non-optimal hypotheses are not strongly penalized during training, if they support artificial data regions which are not consistent w.r.t. the real underlying data distribution.

With our framework ConAD, anomaly detection performance remains competitive or better even with an increasing number of hypotheses available. The discriminator D makes the framework adaptable to the new dataset and less sensitive to the number of hypotheses to be used.

When more hypotheses are used (8), the anomaly detection performance rapidly breaks down. We suggest that the noise is then learned too easily. Consider the extreme case when there are 255 hypotheses available. The Winner-Takes-all-loss will encourage each hypothesis branch to predict a constant image with one value from [0,255]. The discriminator D as a regularizer will try to prevent this effect. That might be a reason why our ConAD has less severe performance breakdown. Our model ConAD is less sensitive to the choice of the hyper-parameter for the number of hypotheses. It also enables better exploitation of the additional expressive power provided by the MHP-technique for new anomaly detection tasks.

## 5 CONCLUSION

In this work, we propose to employ multiple-hypotheses networks for learning data distributions for anomaly detection tasks. Hypotheses are meant to form clusters in the data space and can easily capture model uncertainty not encoded by the latent code. multiple-hypotheses networks can provide a more fine-grained description of the data distribution and therefore enable also a more fine-grained anomaly detection. Furthermore, to reduce support of artificial data modes by hypotheses learning, we propose using a discriminator D as a critic. The combination of multiple-hypotheses learning with D aims to retain the consistency of estimated data modes w.r.t. the real data distribution. Further, D encourage diversity across hypotheses with hypotheses discrimination. Our framework allows the model to identify out-of-distribution samples reliably.

For the anomaly detection task on CIFAR-10, our proposed model results in up to 3.9% points improvement over previously reported results. On a real anomaly detection task, the approach reduces the error of the baseline models from 6.8% to 1.5%.

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

## A   MIXTURE DENSITY NETWORK

The Mixture Density networks predict a data conditional Gaussian mixture model (GMM)in the data space. Conditioning means that each latent vector, i.e., a point on the learned manifold is projected back to a GMM in the data space.

A GMM learns from the following energy function:

$$L_{GMM}(x) = -\log \sum_h \alpha_h \mathcal{N}(x; \mu_h, \sigma_h) \tag{6}$$

Whereby $x$ is the input data, $\mu_h$ and $\sigma_h$ parametrize the $h-th$ Gaussian distribution in the mixture. $\alpha_h$ are the mixing coefficients across the individual mixtures.

Contrary, a Mixture Density network hat multiple output heads (multiple-hypotheses). The framework extends the GMM-learning by the data conditioning as follows:

$$L_{MDN}(x) = E_{z_i \sim q_\phi(z_i|x)}\left[L_{GMM}(x|z_i)\right] \tag{7}$$

whereby $q_\phi$ is a inference network shared by all individual mixtures. $z$ is the latent code. The hypotheses are coupled into forming a likelihood function by the mixing coefficients $\alpha_i$.

## B   MULTIMODAL LEARNING ON THE FLIPPED MOON TOY DATASET

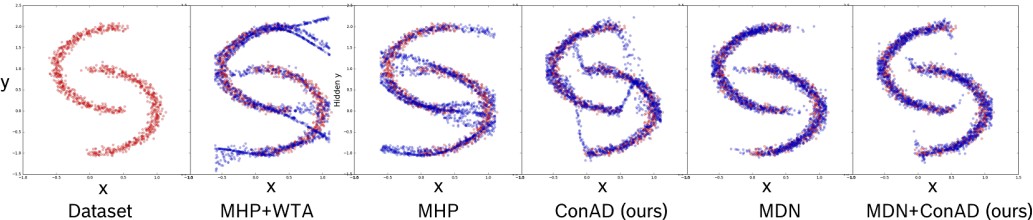

| Dataset | MHP+WTA | MHP | ConAD (ours) | MDN | MDN+ConAD (ours) |

Figure 8: Flipped half-moon dataset: conditional prediction of $y$ based on $x$. Red points are samples from true distribution while blue points represent samples from distributions approximations. Learning with multiple-hypotheses predictions (MHP) loss or MHP + Winner-takes-all (WTA) loss lead to support of artificial data regions. Mixture density networks and our approach ConAD reduces this effect.

Fig. 4 shows the flipped half-moon dataset to demonstrate MHP-learning in contrast to unimodal output distribution learning. In this section, Fig 8 shows a qualitative evaluation of different MHP-techniques. This task is a one-to-many mapping from $x$ to $y$ with a discontinuity at the point $x = 0$ and $x = 0.5$.

When the local density function abruptly ends, MHP-techniques support artificial data regions since they are not penalized for artificial modes by the objective function as discussed before. We refer to this property as an inconsistency concerning the true underlying distribution. In contrast to that, Mixture Density Networks (MDN) and our ConADs approaches reduce the inconsistencies to the minimum.

## C   ONE-TO-MANY MAPPING TASKS REQUIRE MULTI-MODALITY

Consider a simple toy problem with an observable $x$ and hidden $y$ which is to be predicted and expressed by the conditional distribution $p_{true}(y|x)$ such as in Fig. 4. Since the data conditional is multi-modal for some $x$, an uni-modal output distribution cannot fully capture the underlying distribution. Instead, the bias-free solution for the Mean-Squared-Error-minimizer is the empirical mean $\overline{y_{x_i}}$ of $p_{train}(y|x_i)$ on the training set. However, this learned conditional density does not comply with the underlying distribution: sampled data points fall into the low-likelihood regions under $p_{true}(y|x)$. With increasing number of output hypotheses, the data modes could be gradually

captured. For this task, the energy to be minimized is given by the Negative-log-likelihood of the Mixture Density Network (MDN) App. A under a Gaussian Mixture with hypotheses h in Eq. 9 :

$$E_{MDN}(\Theta) = -\log L(\Theta|X;Y) = -\log p_{GMM}(Y|X,\Theta) = -\sum_i \sum_h \log \alpha_h p_{\theta_h}(y_i|x_i) \quad (8)$$

with

$$p_{\theta_h}(y_i|x_i, \theta_h) = \frac{1}{\sqrt{2\pi}\sigma_h} \exp -\frac{(y_i - \mu_h)^2}{2\sigma_h^2} \quad (9)$$

## D   LEMMA 4.1

Given a sufficient number of hypotheses H', an optimal solution $\Theta^*$ for $E_{WTA}(\Theta^*)$ is not unique (permutation is excluded). There exists a $\Theta'$ with $E_{WTA}(\Theta^*) = E_{WTA}(\Theta')$ which is not consistent w.r.t. the underlying output distribution $p_{train}(y_i|x_i)$.

*Proof.* : Suppose $c$ is the maximal modes count of the dataset sampled from the real underlying conditional output distribution $p(y_i|x_i)$. Since $|\{(x_i, y_i)\}| < \infty \to c < \infty$.

Suppose $H = c$, then a trivial optimal solution for $E_{WTA}(\Theta_H)$ is found by centering each hypothesis $\mu_{ik}$ at a different empirical data point $k$ $y_{ik} \sim (y_i, x_i)$ and $\sigma_{ik} \mapsto 0$. In this case $\lim_{\sigma_{ik} \mapsto 0; \forall i,k} E_{WTA}(\widehat{\Theta}_H) = 0$.

Suppose $H' > c$, then a solution $\widehat{\Theta}_{H'}$ can be formulated s.t.: $E(\widehat{\Theta}_H) = E(\widehat{\Theta}_{H'})$.

Let $\widehat{\Theta}_{H'} = \widehat{\Theta}_H \cup \widehat{\Theta}_{H+1...H'} = \widehat{\Theta}_H \cup \{\theta_{h+1}...\theta_{h'}\}$ for some **random** $\widehat{\Theta}_{H+1...H'}$. Due to randomness and without loss of generality, one can assume that $\forall(x_i, y_i), \forall \theta_i \in \Theta_{H+1...H'}$, $\theta_i$ is not the optimal hypothesis for any training point $(x_i, y_i) \in D_{train}$.

In this case due to the winner-takes-all energy formulation we have:

$$E_{WTA}(\widehat{\theta}_{H'}) = -\sum_i \max_{1 \le h \le H'} \log p_{\theta_h}(y_i|x_i) = -\sum_i \max_{1 \le h \le H} \log p_{\theta_h}(y_i|x_i) = E_{WTA}(\widehat{\theta}_H) \quad (10)$$

So $\widehat{\Theta}_H$ and $\widehat{\Theta}_{H'}$ with $H' > H$ are both solutions to the loss formulation and share the same energy level. The extended hypotheses can support arbitrary artificial data regions without being penalized. □

## E   LEMMA 4.2

$$E_{MHP}(\Theta) = -\sum_i \sum_h \log (p_{\theta_h}(y_i|x_i)) * \begin{cases} 1 - \epsilon, p_{\theta_h}(y_i|x_i) \ge p_{\theta_k}(y_i|x_i), \forall k \\ \frac{\epsilon}{H-1}, \text{else} \end{cases} \quad (11)$$

Whereby $x_i, y_i$ is corresponding input-output pairs from the training dataset, $1 \le h \le H$ is a hypothesis branch, which is generated by a parametrized neural network with the parameter set $\theta_h$. Furthermore, $\epsilon$ is a hyperparameter used to distribute the learning signal to the non-optimal hypotheses. $\Theta$ is the collection of all $\theta_h$.

**Lemma E.1.** *Similar to Lemma D, minimizing $E_{MHP}$ in Eq. 11 might also lead to an inconsistent approximation of the real underlying output distribution.*

*Proof.* First, note that $0 \le \epsilon \le \frac{H-1}{H}$, since $\epsilon < 0$ would push away non-locally optimal hypotheses from the empirical solution, $\epsilon > \frac{H-1}{H}$ would penalize the best hypothesis more than others. Both are undesired properties of MHP-learning. First consider the case where $\epsilon \mapsto \frac{H-1}{H}$ :

$$\lim_{\epsilon \mapsto \frac{H-1}{H}} E_{MHP}(\Theta) = \sum_i \sum_h \log (p_{\theta_h}(y_i|x_i)) * \frac{1}{H} \quad (12)$$

$$= \frac{1}{H} \sum_h \left( \sum_i \log\left(p_{\theta_h}(y_i|x_i)\right) \right)$$

$$= \frac{1}{H} \sum_h E_{\theta_h}$$

$\forall \theta_h$ and training data points $(x_i, y_{ik})$ the optimal least-squares solution is the mean, therefore we have:

$$\theta_h^*(y_i|x_i) = E_{y_{ik} \sim p(y|x_i)}[y_i]$$

$$= \frac{1}{l} \sum_{i=1}^{l} y_i; y_{ik} \sim p(y_i|x_i)$$

In this case, all hypotheses are optimized independently and converge to the same solution similar to a single-hypothesis approach. The resulting distribution is inconsistent w.r.t the real output distribution (see Fig. 4 for an example).

Now consider $\epsilon \mapsto 1$:

$$\lim_{\epsilon \mapsto 1} E_{MHP}(\Theta) = -\sum_i \sum_h \log\left(p_{\theta_h}(y_i|x_i)\right) * \begin{cases} 1; \text{if } \theta_h \text{ is best hypothesis} \\ 0; \text{else} \end{cases}$$

$$= -\sum_i \max_{1 \le h \le H'} \log p_{\theta_h}(y_i|x_i)$$

$$= E_{WTA}(\Theta)$$

In this case $E_{MHP}$ shares the same inconsistency property with $E_{WTA}$. Consequently, choosing $\epsilon \in [0, \frac{H-1}{H}]$ only smoothes the penalty on suboptimal hypotheses. The risk remains that distributions induced by non-optimal hypotheses are beyond the real modes of the underlying distribution.

□

## F    EXPERIMENTS DETAILS

**Network architecture**    The networks are following DCGAN (Radford et al., 2015a) but only scaled down to support low-resolution of CIFAR-10. Concretely, the decoder (generator) only uses deconvolutional layers. Throughout the network, leaky-relu units are employed. The framework is implemented in Lasagne (Dieleman et al., 2015) /Theano (Bergstra et al., 2010; Bastien et al., 2012).

Hypotheses branches are represented as decoder networks heads. Each hypothesis predicts one Gaussian distribution with diagonal co-variance $\Sigma$ and mean . The winner-takes-all loss operates on pixel-level,i.e., for each predicted pixel, there is a single winner across hypotheses. The best-combined-reconstructions is the combination of winning hypotheses on pixel-level.

**Training**    We feed the fake images to the discriminator D, consisting of 4 batches:

- real n real images
- fake: n random hypotheses from image reconstructions hypotheses branches
- fake: n best-combined (based on winner hypotheses) reconstructions
- fake: n random sampled images from latent prior $\mathcal{N}(0, 1)$

The batch-size $n$ was set to 64 each on CIFAR-10, 32 on Metal Anomaly. The training was performed with Adam (Kingma & Ba, 2014) with a learning rate of 0.001. Per discriminator training, the generator is trained at most five epochs to balance both players.

| CIFAR-10 | 0 | 1 | 2 | 3 | 4 | 5 | 6 | 7 | 8 | 9 | Mean |
|---|---|---|---|---|---|---|---|---|---|---|---|
| KDE-PCA | .705 | .493 | **.734** | .522 | .691 | .439 | .771 | .458 | .595 | .490 | .590 |
| KDE-Alexnet | .559 | .487 | .582 | .531 | .651 | .551 | .613 | .593 | .600 | .529 | .570 |
| OC-SVM-PCA | .666 | .473 | .675 | .530 | .827 | .438 | **.787** | .532 | .720 | .453 | .610 |
| OC-SVM-Alexnet | .594 | .540 | .588 | .575 | .753 | .558 | .692 | .547 | .630 | .530 | .601 |
| IF | .630 | .379 | .630 | .408 | **.764** | .514 | .666 | .480 | .651 | .459 | .558 |
| GMM | .709 | .443 | .697 | .445 | **.761** | .505 | .766 | .496 | .646 | .384 | .585 |
| AnoGAN | .610 | .565 | .648 | .528 | .670 | .592 | .625 | .576 | .723 | .582 | .612 |
| ADGAN | .632 | .529 | .580 | **.606** | .607 | **.659** | .611 | .630 | .744 | .644 | .62 |
| VAE | .771 | .467 | .684 | .538 | .71 | .542 | .642 | .512 | **.765** | .467 | .610 |
| VAEGAN | .762 | .469 | **.697** | .520 | .756 | .536 | .588 | .554 | .754 | .460 | .609 |
| OC-D-SVDD | .617 | .659 | .508 | .591 | .609 | .657 | .677 | .673 | .759 | .731 | .632 |
| MDN-2 | .761 | .469 | .687 | .538 | .704 | .538 | .632 | .523 | **.768** | .467 | .609 |
| MDN-4 | .769 | .468 | .686 | .535 | .693 | .544 | .635 | .541 | .76 | .469 | .610 |
| MDN-8 | .762 | .469 | .686 | .533 | .704 | .547 | .633 | .53 | .763 | .473 | .61 |
| MDN-16 | .762 | .479 | .682 | .528 | .701 | .54 | .635 | .529 | .764 | .469 | .609 |
| MHP-WTA-2 | .773 | .516 | .68 | .552 | .695 | .543 | .643 | .555 | .76 | .512 | .622 |
| MHP-WTA-4 | **.778** | .539 | .651 | .567 | .66 | .542 | .635 | .563 | .752 | .541 | .622 |
| MHP-WTA-8 | .761 | .56 | .627 | .588 | .626 | .553 | .614 | .578 | .743 | .548 | .619 |
| MHP-WTA-16 | .757 | .567 | .609 | .598 | .627 | .56 | .61 | .568 | .738 | .573 | .62 |
| MHP-2 | .755 | .499 | .676 | .546 | .693 | .543 | .636 | .577 | .764 | .508 | .619 |
| MHP-4 | .752 | .51 | .66 | .568 | .677 | .551 | .644 | .56 | .764 | .51 | .619 |
| MHP-8 | .757 | .54 | .652 | .576 | .648 | .554 | .625 | .547 | .759 | .53 | .618 |
| MHP-16 | .758 | .539 | .641 | .585 | .646 | .552 | .623 | .545 | .759 | .532 | .617 |
| MDN+ConAD-2 | .746 | .489 | .686 | .521 | .711 | .525 | .668 | .577 | .765 | .481 | .616 |
| MDN+ConAD-4 | .762 | .504 | .69 | .524 | .716 | .532 | .659 | .583 | .753 | .489 | .621 |
| MDN+ConAD-8 | .774 | .483 | .693 | .531 | .722 | .537 | .679 | .54 | .76 | .519 | .623 |
| MDN+ConAD-16 | .736 | .469 | .694 | .522 | .753 | .541 | .657 | .568 | .753 | .454 | .614 |
| ConAD - 2 (ours) | .773 | .600 | .666 | .562 | .694 | .561 | .706 | .630 | .748 | .499 | .643 |
| ConAD - 4 (ours) | **.776** | .525 | .663 | .570 | .687 | .541 | **.801** | .548 | .741 | .539 | .639 |
| ConAD - 8 (ours) | .774 | **.652** | .648 | .601 | .670 | .579 | .725 | **.662** | .748 | **.660** | **.671** |
| ConAD - 16 (ours) | .772 | **.631** | .631 | **.615** | .633 | **.588** | .691 | **.640** | .755 | **.637** | **.659** |

Table 4: CIFAR-10 anomaly detection: AUROC-performance of different approaches. The column indicates which class was used as in-class data for distribution learning. Note that random performance is at 50% and higher scores are better. Top-2-methods are marked. Our ConAD approach outperforms traditional methods and vanilla MHP-approaches significantly and can benefit from an increasing number of hypotheses. Furthermore, Mixture Density Networks perform similarly to uni-modal output distributions of VAEs.

# G  CIFAR-RESULTS

# H  METAL ANOMALY DATASET

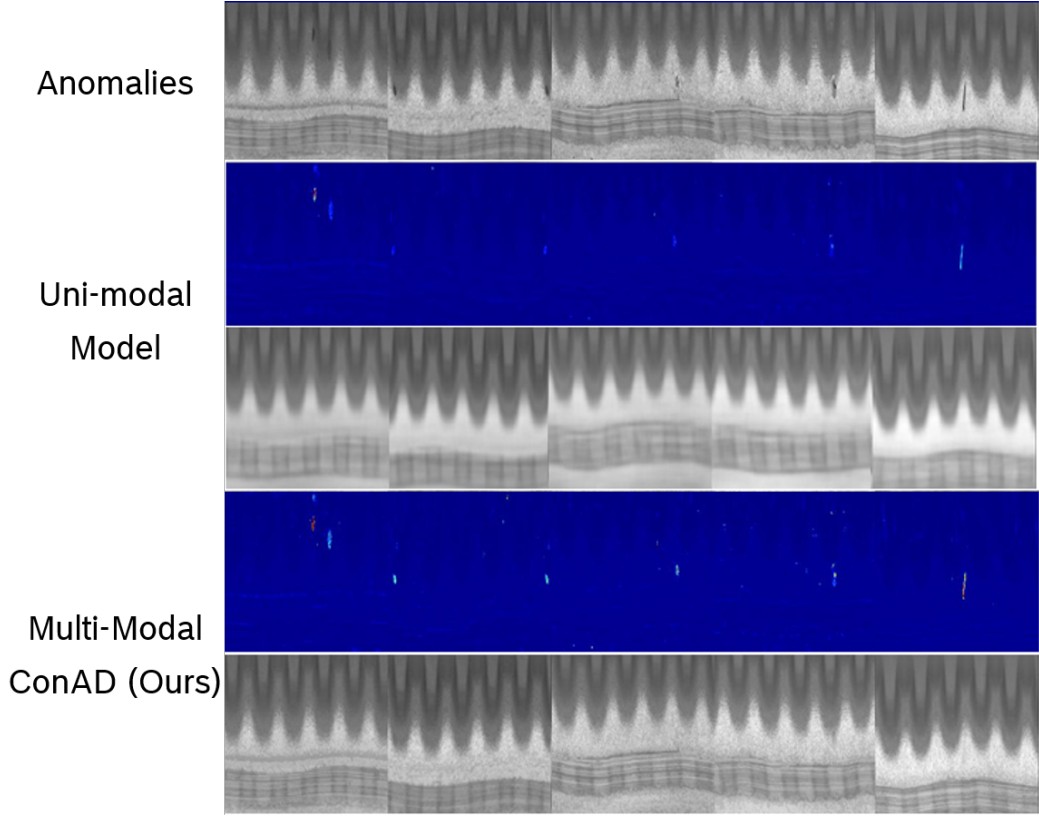

Figure 9: Metal Anomaly dataset. Image reconstructions: reconstructions from uni-modal models are blurry at convergence. Using our ConAD-approach (last two rows), the maximally consistent reconstruction is closer to the original image, capturing many more details needed to differentiate between normal data noise and real anomalies, such as black spots or scratches. The likelihood maximizer in the hypotheses space is much closer to the original and also more realistic. The residuals are significantly clearer for our ConAD-method.

| Model | NLL-All-pixels | 10% abnormal pixels | 1%-abnormal pixels |
|---|---|---|---|
| VAE | .795 | .942 | .977 |
| VAEGAN (1-hyp) | .782 | .936 | .978 |
| MDN-2 | .746 | .900 | .970 |
| MDN-4 | .765 | .910 | .960 |
| MDN-8 | .743 | .916 | .975 |
| MDN+ConAD-2 (ours) | .810 | .942 | .966 |
| MDN+ConAD-4 (ours) | .781 | .913 | .951 |
| MDN+ConAD-8 (ours) | .810 | .943 | .978 |
| MHP-2 | .876 | .980 | **.993** |
| MHP-4 | .834 | .970 | .990 |
| MHP-8 | .793 | .950 | .984 |
| MHP-WTA-2 | .851 | .980 | .990 |
| MHP-WTA-4 | **.878** | .980 | .990 |
| MHP-WTA-8 | .800 | .946 | .981 |
| ConAD-2 (ours) | .867 | **.985** | **.992** |
| ConAD-4 (ours) | .812 | .977 | .990 |
| ConAD-8 (ours) | .817 | .965 | .987 |

Table 5: Anomaly detection performance on the Metal Anomaly dataset, measured in AUROC, showing how different multiple hypothesis approaches perform with increasing number of hypotheses. Vanilla single-hypothesis approaches such as VAE and VAE+GAN under-perform on this task. Even with more sophisticated multi-modal output distribution capacity (MDN), the discriminability is not improved. The integration of MDN into the GAN-framework only slightly improves the results. On the other hand, all other MHP-approaches perform similarly well with $> 99\%$ AUROC (at 1% of most abnormal pixels considered), which indicates that the task has become easily solvable for these methods.

