# OpenReview forum: "Consistency-based anomaly detection with adaptive multiple-hypotheses predictions"
_ICLR.cc/2019/Conference_

### Official Review · AnonReviewer2 · 2018-10-29
**Confusing paper, some good ideas**

**Rating:** 5
**Confidence:** 4

**Review:**

The paper proposes a new method for anomaly detection using deep learning. It works as follows.

The method is based on the recent Multiple-Hypotheses predictions (MHP) model, the impact of which is yet unclear/questionable. The idea in MHP is to represent the data using multiple models. Depending on the part of the space where an instance falls, a different model is active. In this paper this is realized using VAEs. The details are unclear (the paper is poorly written and lacks some detailed explainations), but I am assuming that for each hypothesis (ie region of the space) different en- and decoder parameters are learned (sharing the same variational prior??). The authors mention that below this final layer all hypothesis share the same network parameters. An adversarial loss is added to the model (how that is done is not described; the relevant equation (5) uses L_hyp which is not defined) to avoid the mode collapse.

What is interesting about the paper:
- First of all, pushing the the MHP framework towards AD could be relevant by its own right for a very small subcommunity that is interested in this method
- The idea of using the adv loss for avoiding mode collapse can be useful in other settings; this is def a that I learned from the paper
- The method might actually work rather well in practice

Votum. As outlined above, the paper makes some rather interesting points, but is not well written and lacks some details. I am not entirely convinced that AD and MHP is a killer combination, but the experimental results are ok, nothing to complain here (except the usual bla: make it larger, more, etc), but honestly they really fine (maybe compare also again against more related work, e.g., Ruff et al ICML 2018).

---

> ### Author Response · Authors · 2018-11-23
> **Main revision to clarify motivation and contribution**
>
> Thank you for your helpful comments. We highly appreciate your feedback to improve our paper.
>
> **General remarks***:
>     - To reduce the confusion about the motivation, the abstract and motivation have been rewritten and restructured.
>     - We increase the number of illustrative figures from 2 to 7 in the main paper.
>     - We update the reference to more recent related work (Ruff et al ICML-2018). Hence Our improvement on CIFAR-10 over SOTA is now 3.9% points instead of 5% point.
> ********Comments******************
>
> Confusing paper, some good ideas
>
> 1. The details are unclear (the paper is poorly written and lacks some detailed explainations)
>     - We added details for training & architecture in the appendix.
>     - The motivation in the introduction is rewritten and hopefully becomes  clearer.
> 2. I am assuming that for each hypothesis (ie region of the space) different en- and decoder parameters are learned (sharing the same variational prior??).
>     - Details are shown in an additional figure (7).
>     - The multiple-hypotheses network shares all layers but the last layer. That means, only the decoder part is changed, to have multiple heads.
>
> 3. An adversarial loss is added to the model (how that is done is not described; the relevant equation (5) uses L_hyp which is not defined) to avoid the mode collapse.
>     - L_{hyps} is a typo, it should be L_{WTA}
>     - How this is learned is described in Figure 7: ConAD: our multiple-hypotheses autoencoder and with the training discriminator train
>     - In short:
>         We feed the fake images to the discriminator D, consisting of 4 batches:
>             - real  32 real images
>             - fake: 32 random hypotheses from image reconstructions hypotheses branches
>             - fake: 32 best-combined (based on  winner hypotheses) reconstructions
>             - fake: 32 random sampled images from  latent prior N(0,1)
>
> 4. The idea of using the adv loss for avoiding mode collapse can be useful in other settings.
>     - We also expect it to be beneficial in normal use-cases of MHP-learning.  Instead of designing some additional losses to make the hypotheses  meaningful, the work can be shifted to the discriminator.
>
> 5. [...] compare also again against more related work, e.g., Ruff et al ICML 2018).
>     - Thank you for pointing this out. We have added their results for comparison.
>
> 99. [...] not well written and lacks some details.
>     - we have worked on the rewriting to make the motivation and idea clearer.
>     Details are added in the paper as well as attachment.

---

### Official Review · AnonReviewer3 · 2018-11-02
**Technique to make GANs more robust by adding MHP-based regularization so that detection of out-of-distribution samples can be improved**

**Rating:** 5
**Confidence:** 3

**Review:**

Summary
-------
The paper proposes a technique to make generative models more robust by making them consistent with the local density. It is hypothesized that robust model will be able to detect out-of-distribution samples better and improve anomaly detection.

Main comments
-------------

1. The proposed technique adds additional regularizers to the GAN loss that, in effect, state that the best hypothesis under a WTA strategy should have a high likelihood under the discriminator 'D'. This is an interesting idea and certainly a reasonable thing to try. As stated in the abstract, the generative models are inefficient; it is likely that additional structure enforced by the regularizer helps in improving the efficiency.

2. The objective in GANs is to infer the underlying distribution correctly and so far it has been found that their accuracy is heavily dependent on both the architecture as well as the computational complexity (they may improve with more training, but maybe not consistently). Therefore, it becomes hard to compare the three architectures in Figure 2 since they are all different. A more rigorous comparison would try to keep as many pieces of the architecture the same as possible so that ConAD can be compared with 'all other things being same'. Some experiments seem to follow this idea such as 'MDN+ConAD-{2, 4, 8, 16}' in Table 2. But in these experiments the addition of ConAD offers a mild improvement and even degrades for the maximum number of hypothesis (i.e., 16).

3. Page 2, para 2, last two lines: "For simplicity, imagine an ... the real distribution."

The argument is not clear. It seems too trivial and almost like a straw man argument.

4. Page 4: "In anomaly detection, this is difficult since there is no anomalous data point contained in the training dataset."

This is not true in real-world applications where most data is contaminated with anomalies. This is part of the challenge in anomaly detection.

The above also applies to the following on page 6: "During model training, only data from the normal data class is used..."

5. Page 5: "...D minimizes Eq. 3": Should be 'maximizes' since the reference is to the log likelihood of real data (or, add a negative sign).

6. Eq. 4: The last component should be negative since we trying to maximize the likelihood of the best hypothesis under WTA (right?).

7. Table 1: The datasets are not real anomaly detection datasets (too high proportion of 'anomalies') Moreover, the number of datasets is insufficient for rigor.

8. Section 5.4: "With our framework ConAD, anomaly detection performance remains competitive or better even with an increasing number of hypotheses available."

Section 6: "... and alleviates performance breakdown when the number of hypotheses is increased."

This is not entirely supported by the results in Tables 2, 3, and also 4 and 5 of supplement. The results for ConAD - {2, 4, 8, 16} are not consistently increasing.

Since experiments are very few (and not real-world for anomaly detection task) because of which the observations cannot be generalized.

9. Page 4 (minor) in two places: "one-to-mapping" -> "one-to-many mapping"

10. Page 5 (minor): "chap. 3" -> "section 3"

---

> ### Author Response · Authors · 2018-11-23
> **Main revision to clarify motivation and contribution (2)**
>
> 6. Eq. 4: The last component should be negative since we trying to maximize the likelihood of the best hypothesis under WTA (right?).
>     - No, we minimize it.
>     - Figure (7) in the appendix shows how the training works. The combined best guess should be detected as fake from the Discriminator. In this way, the generator attempts to make the best guess closer to normal data distribution
>     - The Winner-takes-all loss is one additional likelihood-loss, L_{WTA}, which is a pixel-wise likelihood-loss. The loss only goes to the hypothesis branch which was closest to the input samples.
>
> 99. Since experiments are very few (and not real-world for anomaly detection task) because of which the observations cannot be generalized.
>     - The Metal Anomaly Dataset is a real task from quality control in manufacturing. The goal is to detect anomalies like scratches and dents on a structured metal surface.
>     - On CIFAR-10, results are averaged over 10 x (one vs nine) anomaly detection tasks with multiple runs each.
>     - Our criteria for choosing the datasets are:
>         - Clean data for normal data distribution i.e. no multiple labels, known noise.
>         - Large number of anomalies available to assess the approaches' performance.
>         - Meaningful & hard tasks which have been used by other authors to allow a fair comparison.
>     - Finding a good dataset is a common problem for the semi-supervised anomaly detection community. For example, Ruff et al (ICML-18) only perform evaluation on "artificial" anomaly detection set (CIFAR-10). Our work however, extends the analysis to Metal Anomaly, a real anomaly detection task.
> *****Minor*******:
> 5. Page 5: "...D minimizes Eq. 3": Should be 'maximizes' since the reference is to the log likelihood of real data (or, add a negative sign).
> 9. Page 4 (minor) in two places: "one-to-mapping" -> "one-to-many mapping"
> 10. Page 5 (minor): "chap. 3" -> "section 3"
>     - Thank you for catching these mistakes. We have fixed these issues and some other we could spot.

---

> ### Author Response · Authors · 2018-11-23
> **Main revision to clarify motivation and contribution (1)**
>
> Thank you for your helpful comments. We highly appreciate your feedback to improve our paper.
>
> **General remarks***:
>     - To reduce the confusion about the motivation, the abstract and motivation have been rewritten and restructured.
>     - We increase the number of illustrative figures from 2 to 7 in the main paper.
>     - We update the reference to more recent related work (Ruff et al ICML 2018, as pointed out by reviewer2). Hence our improvement on CIFAR-10 over SOTA is now 3.9% points instead of 5% point.
> ********Comments******************
>
> 1. [...] ConAD offers a mild improvement and even degrades for the maximum number of hypothesis (i.e., 16).
>     (See 8.)
> 2. ConAD can be compared with 'all other things being same'.
>     - Fig. 2 only illustrates the difference between single- and multiple-hypotheses networks. In our training, the networks are kept as similar as possible.
>     - Our ablation experiment is somewhat implicit as follows:
>         - Without the discriminator, the model becomes MHP-WTA
>         - Without multiple hypotheses, it becomes VAE-GAN
>         - Without multiple-hypotheses & discriminator, it becomes VAE
>         - Without pixel-wise likelihood-learning, anomaly detection is random (not reported). VAE-GAN training was also reported to require pixel-wise likelihood, in addition to perceptual metric from the discriminator.
>     - We extended the appendix to make the network architectures and training clearer.
>
>
> 3. Page 2, para 2, last two lines: "For simplicity, imagine an ... the real distribution. [...] too trivial and almost like a straw man argument.
>     - Due to its perception as a triviality, we have removed this argument and restructured the introduction to communicate our motivations ideas more clearly.
> 4. [...] in real-world applications [...] most data is contaminated with anomalies [...]
>     - In many cases, the data is contaminated with anomalies. This is certainly an important challenge and belongs to unsupervised anomaly detection
>     - However, One-class-learning tasks are also very important (and frequent) in pratice. In this case, normal data can easily be collected and labeled. For these scenarios, the main effort comes from the data collection process, not the labeling.
>     - For example: consider setting up a new production line in manufacturing and setting up an anomaly detection model for quality control. Gathering many normal samples and to label them as such is feasible. However, to get one single real anomaly, one has to wait for a long time. This happens because anomalies are extremely rare  and do not need to appear right in the beginning.
>     - Another sample: Disease detection: Some diseases are extremely rare, finding the patients infected with exactly this disease is difficult and hence expensive. Contrary, data acquisition from healthy patients is quite easy and effortless.
> 7. Table 1: The datasets are not real anomaly detection datasets (too high proportion of 'anomalies')[...]
>     - It is possible to reduce the number of anomaly samples to resemble realistic scenarios. However, the performance measures are then statistically less certain.  Furthermore, The Area-under-curve-score (AUROC) is a  class-frequency-normalized metric, i.e. imbalance between anomaly class & normal class is no problem.
>     - Note that the anomalies are only used for testing anomaly detection performance. They are not used for training.
>     - The Metal Anomaly dataset is a real anomaly detection set, where abnormality could occur on the surface. In contrast, CIFAR-10 serves as a benchmark to compare with other works (Deecke et al 2018)
>
> 8. Section 5.4: "With our framework ConAD, anomaly detection performance remains competitive or better even with an increasing number of hypotheses available." Section 6: "... and alleviates performance breakdown when the number of hypotheses is increased.". The results for ConAD - {2, 4, 8, 16} are not consistently increasing.
>     - we added some comments in the experiments part & training details in the attachment
>     - In short: Too many hypotheses also capture much irrelevant noise.
>        Extreme case: When there are 255 hypotheses available:
>        The Winner-Takes-all-loss will encourage each hypothesis branch to predict a constant image with one value from [0,255]. The discriminator D as a regularizer will try to prevent this effect. That might be a reason why our ConAD has less severe performance breakdown, choosing the hyperparameter: hypotheses branches less sensitive.

---

### Official Review · AnonReviewer1 · 2018-11-05
**Does not read well, no clear motivation**

**Rating:** 4
**Confidence:** 4

**Review:**

This paper proposes an anomaly detection system by proposing the combination of multiple-hypotheses approach with variational autoencoders, and using a discriminator to prevent either head of the model to produce modes that are not part of the data.

The combination between multiple-hypotheses approach with variational autoencoders seems rather artificial to me. Why do we need to parameterized a fixed set of hypothesis if we can generate as many outputs as we want just by sample several times from the prior of the VAE? Maybe I am missing something, which brings me to the following point.

The paper is difficult to read: the motivation is not well explained, the link between anomaly detection and multiple-hypothesis methods (both in the title of the paper) is not clear. The approach seems to build on top of Breunig et al. (2000), unfortunately this paper is not well described, e.g. what does it mean global neighborhood?
There are many other sentences in the paper that I find difficult to understand, for example:
"Lfake itself consists of assessment for noise- (xˆz∼N(0,1)) and data-conditioned (xˆz∼N(µz|x,Σz|x)) hypotheses and the best guess given by the WTA objective."

On top of that there are many other elements in the paper hampering the comprehension of the reader. For example:
WTA is used without being defined before (winner takes all)
one-to-mapping --> one-to-one mapping?
L_[Hyps] is the same as L_[WTA]?
MDN is not defined until Sec. 5, and doing so without giving any description about it.
Table 3 is never referred to.
Is Table 5 reporting results on the Metal anomaly dataset? If so please mention it in the caption.

In the experiments it is difficult to see which parts of the models make the main difference. For example, it would be interesting to have an ablation experiment assessing the importance of the discriminator.

---

> ### Author Response · Authors · 2018-11-23
> **Main revision to clarify motivation and contribution (2)**
>
> 4. There are many other sentences in the paper that I find difficult to understand, for example [...]:
>     - We have changed many parts of the paper to make it more clear  & simple to read.
> 5. Which parts of the models make the main difference. [...] ablation experiment assessing the importance of the discriminator
>     - Our ablation experiment is somewhat implicit as follows:
>         - Without the discriminator, the model becomes MHP-WTA
>         - Without multiple hypotheses, it becomes VAE-GAN
>         - Without multiple-hypotheses & discriminator, it becomes VAE
>         - Without pixel-wise likelihood-learning, anomaly detection is random (not reported). VAE-GAN training was also reported to require pixel-wise likelihood, in addition to perceptual metric from the discriminator.
> 6.MDN is not defined until Sec. 5, and doing so without giving any description about it.
>         - we added a more formal definition about MDN in the Appendix. Furthermore, we explain the idea behind MDN more deeply in the introduction.
> ***Minor***
> - WTA is used without being defined before (winner takes all)
> - L_[Hyps] is the same as L_[WTA]?
> - one-to-mapping --> one-to-one mapping?
> - Table 3 is never referred to.
> -  Is Table 5 reporting results on the Metal anomaly dataset? If so please mention it in the caption.
> -  "Lfake itself consists of assessment for noise- (xˆz∼N(0,1)) and data-conditioned (xˆz∼N(µz|x,Σz|x)) hypotheses and the best guess given by the WTA objective."
>     -Figure 7 illustrates how samples are fed into the discriminator.  Samples labeled as fake are: randomly-sampled images $\hat{x}_{z\sim \mathcal{N}(0,1)}$, data reconstruction defined by individual hypotheses $\hat{x}_{z\sim \mathcal{N}(\mu_{z|x},\Sigma_{z|x})}$, the best combination of hypotheses according to the Winner-takes-all-loss $\hat{x}_{\text{best\_guess}}$.
>
>
>     - Thank you for pointing out. We fixed this and other points we could spot.

---

> > ### Comment · AnonReviewer1 · 2018-12-14
> > **Thanks for the response**
> >
> > Thanks to the authors for their response, and for updating the paper accordingly. The motivation is now somewhat clearer, but I would still recommend resubmitting after some re-work. For example, if the focus of the paper is on detecting anomalies by means of the predicted log-likelihood, why is there so much focus on learning good reconstructions? Also I assume that the VAE employed in the experiments has as many parameters as the VAE part of the proposed approach. However the proposed approach has way more parameters (multiple heads plus the discriminator). Maybe a more powerful VAE is able to learn better log-likelihoods?

---

> ### Author Response · Authors · 2018-11-23
> **Main revision to clarify motivation and contribution (1)**
>
> Dear reviewer,
>
> Thank you for your helpful comments. We highly appreciate your feedback to improve our paper.
>
> **General remarks***:
>     - To reduce the confusion about the motivation, the abstract and motivation have been rewritten and restructured.
>     - We increase the number of illustrative figures from 2 to 7 in the main paper.
>     - We update the reference to more recent related work (Ruff et al ICML 2018, as pointed out by reviewer2). Hence our improvement on CIFAR-10 over SOTA is now 3.9% points instead of 5% points.
> ***Your comments***
> 1.  Why do we need to parameterize a fixed set of hypotheses if we can generate as many outputs as we want just by sampling several times from the prior of the VAE?
>     - When we sample multiple times from the (data conditioned) prior, we indeed may get many outputs. However, they are not necessary different. In our experience, they are usually very similar and blurry.
>     - A more theoretical perspective: Given the VAE's expressive power, the latent code could capture the real data manifold rather well in theory. In that case, each sampling from the prior leads to one real and sharp image.
>         + But this requires a powerful encoder and/or latent code learning . Recent advances in generative modeling addresses better factorization of latent code [zhao,2017], [Rezende,2015]. Although meaningful, the models are (1) more complex (2) could lead to more data-hungry approaches.
>         + Further, learning from a pixel-wise metric such as likelihood often leads to a blurry images [Ledig -2016]  Longer training helps for training data but reduces likelihood on unseen data. Further, we consider  the blurriness as an expression of model uncertainty. When  reconstructed region is blurry, it means that the network has to regress between different modes (Multi-modality).
>         + Alternative are learning from perceptual metrics  such as GAN and VAE-GAN. But in these approaches, data reconstructions are also very different from input images. see reconstruction of inputs in [Dosovitskiy - 2016, Larsen -2015]
>     ---> a fixed set of hypotheses is an easy way for the network to address multiple modes in the datas pace. In this sense, it's a structured way to express the ambiguity from the latent code to image space.
> 2. The paper is difficult to read: the motivation is not well explained [...]
>     - A hopefully more clear introduction is in the paper.
>     - In short:
>         - Autoencoder-based approaches often produce blurry reconstructions and falsify the reconstruction errors for anomaly detection. Due to likelihood-learning, they regress to the means in the data space.
>         - Better latent code factorization or perceptual metric learning could provide some help but are (1) complex to implement and (2) more data intensive.
>         - We propose a different solution: Let the network express the uncertainty directly with multiple hypotheses. Hence, instead of mean regressions, the hypotheses clusters should be spread out to cover dense data regions.
>
> 3. The approach seems to build on top of Breunig et al. (2000), unfortunately, this paper is not well described, e.g. what does it mean global neighborhood?
>     - Local-outlier-factor computes an outlier score based on its intermediate neighborhood.
>     - It is proportional to: (mean densities of the neighboring points)/(local density around the new point). Higher means more abnormal. So Local-outlier-factor only concentrates on a small neighborhood.
>     - By global neighborhood, we mean that anomaly score computation depends on ALL points in the training data, as in case of a Mixture Density Network, where a Gaussian Mixture is estimated in the image space. When the likelihood is used for anomaly score computation, it's somewhat similar to using weighted distances to ALL points available.

---

### Meta-Review · Area_Chair1 · 2018-12-15
**Interesting approach, but poorly written. Needs to more work before acceptance.**

**Confidence:** 5
**Recommendation:** Reject

**Metareview:**

This paper proposes an anomaly-detection approach by augmenting VAE encoder with a network multiple hypothesis network and then using a discriminator in the decoder to select one of the hypothesis. The idea is interesting although the reviewers found the paper to be poorly written and the approach to be a bit confusing and complicated.

Revisions and rebuttal have certainly helped to improve the quality of the work. However, the reviewers believe that the paper require more work before it can be accepted at ICLR. For this reason, I recommend to reject this paper in its current state.